# Raman Spectroscopy as a Tool to Study the Pathophysiology of Brain Diseases

**DOI:** 10.3390/ijms24032384

**Published:** 2023-01-25

**Authors:** Oihana Terrones, June Olazar-Intxausti, Itxaso Anso, Maier Lorizate, Jon Ander Nieto-Garai, Francesc-Xabier Contreras

**Affiliations:** 1Department of Biochemistry and Molecular Biology, Faculty of Science and Technology, University of the Basque Country (UPV/EHU), Barrio Sarriena s/n, 48940 Leioa, Spain; 2Structural Glycobiology Laboratory, Biocruces Bizkaia Health Research Institute, Cruces University Hospital, 48903 Barakaldo, Spain; 3Instituto Biofisika (UPV/EHU, CSIC), Barrio Sarriena s/n, 48940 Leioa, Spain; 4Ikerbasque, Basque Foundation of Science, 48011 Bilbao, Spain

**Keywords:** Raman spectroscopy, SERS, CARS, brain, traumatic brain injury, cancer, neurodegenerative disease, Alzheimer’s disease, Parkinson’s disease

## Abstract

The Raman phenomenon is based on the spontaneous inelastic scattering of light, which depends on the molecular characteristics of the dispersant. Therefore, Raman spectroscopy and imaging allow us to obtain direct information, in a label-free manner, from the chemical composition of the sample. Since it is well established that the development of many brain diseases is associated with biochemical alterations of the affected tissue, Raman spectroscopy and imaging have emerged as promising tools for the diagnosis of ailments. A combination of Raman spectroscopy and/or imaging with tagged molecules could also help in drug delivery and tracing for treatment of brain diseases. In this review, we first describe the basics of the Raman phenomenon and spectroscopy. Then, we delve into the Raman spectroscopy and imaging modes and the Raman-compatible tags. Finally, we center on the application of Raman in the study, diagnosis, and treatment of brain diseases, by focusing on traumatic brain injury and ischemia, neurodegenerative disorders, and brain cancer.

## 1. Raman Scattering

The Raman phenomenon takes its name from one of its discoverers, C.V. Raman; however, that the wavelength of scattered light shifts when it interacts with matter was independently discovered by C.V. Raman and his student K.S. Krishnan and by L.I. Mandelstam and G.S. Landsberg. Although Mandelstam and Landsberg reported their observations a week earlier, their experiments were limited to crystals and they did not give a comprehensive interpretation of their discovery according to the Physics Nobel Committee. Therefore, only C.V. Raman was awarded in 1930 with the Nobel Prize. C.V. Raman and K.S. Krishnan saw changes in the light’s wavelength when it passed through a transparent liquid, what they called “modified scattering”, which would later be renamed the Raman effect or Raman scattering [1].

After the initial description of this new scattering of the light, the physical phenomena behind the Raman effect were progressively better understood, and it was determined to involve transitions of the vibrational state of molecules. When monochromatic radiation impinges on matter, in any of its physical states, 1 out of 10^4^ photons are spontaneously scattered or dispersed at the same wavelength (or energy) as the incident light. This phenomenon is known as elastic scattering or Rayleigh scattering (Figure 1a). In addition, 1 out of 10^8^ photons is spontaneously scattered inelastically, which means that dispersed photons have either longer (less energy) or shorter (more energy) wavelengths than the incident radiation. This shift in the photon’s wavelength is due to an energy exchange with the matter under study and is related to its molecular composition and structural properties. Scattering involves the transition of the vibrational energy state of a molecule to a higher virtual energy state induced by the interaction with a photon. The virtual state is unstable, and the photon is rapidly re-emitted at a different frequency in the case of inelastic or Raman scattering. Two different Raman scatterings are defined depending on the energy of the re-emitted photon. In the Stokes Raman scattering, the molecule starts at a ground vibrational level. When interacting with the light, energy is transferred to the molecule, which ends in a higher quantized vibrational state, i.e., it gains vibrational energy, while the re-emitted photon has a lower frequency (less energy) than the incident one (Figure 1b). In the anti-Stokes scattering, the molecule starts in a higher vibrational state than in the Stokes mode. When the interaction occurs, an increase in photon energy leaves the molecule in a lower vibrational energy state (Figure 1b). Because—conforming to the Boltzmann distribution—the vast majority of molecules are in the ground vibrational level at room temperature, Stokes scattering is much more frequent than anti-Stokes scattering [2,3,4,5]. The Raman phenomenon must not be confused with infrared (IR) absorption, although both processes imply vibrational energy state transitions. In Raman spectroscopy, the vibrational energy change of a molecule is measured by detecting the scattered photon that gains or loses energy. In contrast, in IR spectroscopy the vibrational energy state is measured by the absorption of the photon by the molecule, which induces the vibrational energy transition (Figure 1c) [6].

The scattering of light in the Raman phenomenon occurs by the formation of oscillating dipoles in the molecules, generated by the oscillating electric field component of the incident electromagnetic radiation. The magnitude of the induced electric dipole moment is proportional to the strength of the applied radiation and the polarizability of the molecule, which refers to the tendency of the electronic cloud to deform when subjected to an external electric field. This distortion from the initial shape generates a charge segregation inducing a dipole momentum, which should not be confused with the molecule’s permanent dipole moment required for the absorption of IR radiation. These differences in the properties of induced dipoles and permanent dipoles make some chemical groups active for IR and not for Raman and vice versa, but coincidentally, both techniques are based on the recording of molecular spectra generated by transitions between quantized vibrational energy levels. Since these levels are characteristic of each molecule, both Raman and IR are valuable tools that provide detailed information about chemical structure and identity. 

Label-free detection of all biomolecules comprises a hallmark advantage of these vibrational spectroscopic techniques. Nevertheless, Raman spectroscopy has the benefit over IR spectroscopy, that in biological samples no interference from the water signal between 300 and 3100 cm^−1^ exists, where most Raman measurements are made, and no special sample preparation is required, facilitating measurements in vivo and aqueous solutions. In addition, IR absorption spectroscopy operates at mid-infrared light, a low frequency radiation resulting in low sample penetration and low spatial resolution. Comparatively, Raman spectroscopy uses near-infrared or visible light, which provides higher sample penetration and spatial resolution.

## 2. Raman Spectroscopy of Biomolecules

As stated above, the Raman scattering induced by a chemical bond depends on its polarizability, and chemical bonds with different properties will scatter the light differently. To define these differences, the Raman shift concept is used. During Raman scattering, the frequency of the re-emitted photon is defined as v_0_ − v_M_ for Stokes, and v_0_ + v_M_ for anti-Stokes, where v_0_ is the frequency (measured as wavenumber) of the incident beam, and v_M_ corresponds to the wavenumber of the vibrational transition. This change in frequency compared to the incident photons (v_0_ − v_M_ or v_0_ + v_M_) is the Raman shift (Δv), which has positive values for Stokes and negative values for anti-Stokes scattering. A specific chemical bond will scatter a photon of a defined frequency with a specific Raman shift. A complex biological sample contains several different chemical bonds, each of which can induce scattering events with different Raman shifts. The collection of the intensities of the Raman shifts derived from the sample comprises its Raman spectrum. Since Stokes and anti-Stokes signals have identical shift magnitudes, and since anti-Stokes signals are significantly less intense than Stokes signals because of their lower probability, commonly only the Stokes region is represented in the Raman spectra.

Virtually all kinds of biomolecules contribute to the Raman spectra of biological samples. Since different biomolecules have specific chemical signatures, several unique vibrational bands can be used for their identification [7,8,9]. The Stokes Raman spectrum of biological molecules is divided into four vibrational frequency regions. The first is the “low-frequency (LF) region”, also known as the terahertz region, which includes frequencies below 300 cm^−1^ and provides key information regarding structural conformation and environmental conditions of biomolecules [10,11]. LF vibrational modes are associated with intra- or intermolecular collective motions [10,12,13]. External perturbations, such as ligand binding, enzymatic activity, electron transfer, or intermolecular interactions, induce fluctuations in these modes providing valuable information about the conformational state of the biomolecule [14,15]. Until recently, LF Raman spectroscopy and microscopy have been limited by the experimental difficulties of measuring at these low frequencies, which include low signal and spatial resolution, and leakage from the input excitation laser. However, recent advances in instrumentation and data analysis have overcome these limitations providing new tools for the study of this spectral region [15,16,17].

The second region is the “fingerprint region” which ranges from 300 cm^−1^ to 1800 cm^−1^ and comprises the most important region for chemical identification. This region has a prominent contribution of signals derived from various biomolecules, including proteins, nucleic acids, lipids, and carbohydrates. Protein contributions rely mainly on the side chain of the aromatic amino acids and the peptide bond. Vibrations of the peptide bond are referred to as Amide modes. Among the nine Amide bands, Amide I (1645–1680 cm^−1^, due to C=O stretching) and III (1225–1280 cm^−1^, due to C-N stretching coupled to N-H bending) are preferentially used to determine the secondary structure of a protein [18,19,20]. Contributions from nucleic acids include both signals from individual bases (600–800 cm^−1^, due to the ring breathing) and from the sugar-phosphate backbone. Additionally, by following the vibrations of phosphodiester bonds in the nucleic acids, the Raman spectra allow the differentiation of A, B, and Z backbone conformations [21]. Vibrations of double bonds (C=N, C=C, and C=O) are also recorded at the fingerprint region (wavenumber window 1500–1800 cm^−1^). The contribution of lipids to the fingerprint region is dominated by the long hydrocarbon chains, providing information about scissoring and twisting of CH_2_ and CH_3_ groups (1300 and 1400–1500 cm^−1^) and C-C stretching (1050–1200 cm^−1^). Additionally, various structures in the lipid headgroups also create several bands in the 710–890 cm^−1^ region [22]. Finally, carbohydrates also contain chemical groups that create bands in the fingerprint region, but the position of those bands varies significantly between molecules and even between monomer and polymer conformations. Nevertheless, some common features such as C-O-C stretching at 850 and 1125 cm^−1^ and CH_3_ rocking at 925 cm^−1^ exist [5,7]. The third region is the “silent region”, between 1800 and 2800 cm^−1^, so called due to the absence of contributions from biomolecules, excluding some exceptions such as alkyne groups. This region tracks signals of specific tags suitable for Raman spectroscopy by avoiding interferences from biological constituents [23,24]. Finally, the “high wavenumber region” above 2800 cm^−1^ is dominated by stretching vibrations of hydrogen bonds (C-H, N-H, O-H) and is suitable for studying lipids and long-chain hydrocarbons [5,22].

### 2.1. Modes of Raman Spectroscopy and Imaging

One of the main advantages of Raman spectroscopy is its spontaneous nature, which allows label-free spectroscopy of biological samples. Nevertheless, as stated above, spontaneous Raman only occurs in approximately 1 out of 10^8^ incident photons, which severely limits the strength of the signal. This is further aggravated by the fact that many biological samples emit fluorescence in the visible range, which can overwhelm the Raman signal. To reduce the contamination from autofluorescence, long wavelength lasers can be used, with the counterpart decrease in the Raman signal [25]. Nowadays, most Raman imaging is carried out between 700 and 900 nm, where a compromise between the autofluorescence and Raman signals can be found. Even so, long acquisition times are required to detect sufficient photons and obtain an acceptable signal to noise. Acquiring sufficient photons to measure a single Raman spectrum requires around 0.5 s in fast systems, implying that a 512 × 512 pixel Raman image would take 36 h to acquire by point-scanning [5]. Several Raman imaging modes and techniques have been developed to overcome this limitation, which can be divided into two main strategies: increasing imaging acquisition speed, and increasing signal strength.

In the first strategy, the image acquisition set-up is modified to increase imaging acquisition speed so that, even with a low Raman signals, images can be acquired in a shorter time. This is usually achieved by departing from the point-scanning approach and trying to acquire several pixels simultaneously. The main approaches that have been developed are line- and slit-scanning [26,27], wide-field imaging with light-sheet microscopy [28,29], and multi-focal imaging [30]. Comprehensive reviews on the characteristics and comparative advantages of these techniques can be found elsewhere [31,32]. These approaches generally reach acquisition speeds 20 times faster than point-scanning imaging, and even though they are not fast enough for live cell imaging, they constitute a suitable alternative for many other applications of Raman spectroscopy.

In the second strategy, the strength of the Raman signal is increased by using different Raman modes, which in turn allows for shorter acquisition times. In the study of brain physiopathology, three main modes are commonly applied to increase the signal strength compared to spontaneous Raman: non-linear Raman scattering techniques, such as Stimulated Raman Scattering (SRS) and Coherent anti-Stokes Raman Scattering (CARS), and Surface-Enhanced Raman Scattering (SERS). 

In the non-linear modes of Raman scattering, a specific vibrational transition is stimulated by the use of multiple lasers, thus increasing the strength of the signal. An in-depth explanation of the physical phenomena behind these techniques is out of the scope of this review and can be found elsewhere [5,33]. Briefly, in SRS, the sample is irradiated with a “pump” laser as in spontaneous Raman, combined with a lower frequency “Stokes” laser. The frequency of the Stokes laser is selected so that the difference in energy between both lasers (Δv) resembles that of a specific vibrational transition, which enhances the occurrence of that transition and increases its signal (Figure 2). For each pump and Stokes frequency combination, a narrowband measurement of a single vibrational peak is obtained. Broadband or hyperspectral measurements are obtained by locking the frequency of one of the lasers and varying the frequency of the other, so the whole range of vibrational transitions is scanned and detected. The increase in signal strength has allowed the development of video-rate imaging of up to 25 fps for a 512 × 512 pixel image [34]. Additionally, in SRS the signal scales linearly with the concentration of the sampled molecules, allowing for quantitative imaging.

CARS is also a non-linear multi-photon technique where the sample is irradiated by pump and Stokes short-pulse lasers. As in SRS, these lasers are tuned so that their energy difference coincides with the energy of a specific vibrational transition of the target molecule, inducing a specific coherent vibration at an energy level higher than the ground state. These vibrating molecules are probed by a third “probe” laser, usually with the same frequency as the pump laser, so that they return to the ground state and produce an anti-Stokes signal of higher frequency than the probe laser (Figure 2). By fixing the wavelength of the pump laser and varying the frequency of the Stokes beam, broadband measurements can be obtained as in SRS [35,36,37]. CARS achieves a 1000-fold increase in signal strength [38], and because the scattered light is blue-shifted, it is not subject to interference from autofluorescence. As in SRS, the increase in signal strength allows for shorter acquisition times, allowing video-rate imaging of up to 20 fps [39]. Unlike SRS, CARS signal is non-linearly dependent on concentration, so quantitative imaging is not straightforward.

The third signal-enhancing technique, SERS, relies on modifying the sample to enhance the signal. In SERS, nanoparticles of metals such as gold and silver are used, which produce strong electromagnetic fields on their surface when struck by the incident light, enhancing the Raman signal of the target molecules. The physical phenomena behind this process are not completely understood, but it has been well established that with SERS the signal can be increased up to 10^14^–10^15^ times, allowing the detection of even single molecules [40,41]. Because the metal surface provides the enhancement, the molecule of interest must interact with the metal to be detected. Although this limits the application of the technique, it allows for selective visualization by using nanoparticles designed to target specific organelles or molecules [42,43,44,45]. 

### 2.2. Raman Tags

The main advantage of Raman spectroscopy in its different imaging modes is that it can be performed label-free, as it measures the inherent vibration of chemical bonds in biomolecules. The advantage of this label-free approach is obvious, as it provides molecular level information without requiring extensive sample preparation or modification. Label-free Raman spectroscopy has been used to study proteins, lipids, nucleotides, and different bioactive molecules in various biological samples [46,47,48,49,50,51,52,53]. In principle, because of the ubiquity of these bonds, proteins, lipids, nucleotides, carbohydrates, and other biomolecules can be visualized simultaneously. Nevertheless, in practice, all of the molecules containing the same bond will produce overlapping spectra making it extremely challenging to attribute the signal from a specific chemical bond to a unique type of biomolecule, severely limiting detection specificity. To overcome this problem, different Raman tags have been developed. These tags are small functional groups or isotopes that vibrate in the “silent region” between 1800 cm^−1^ and 2800 cm^−1^ in which no naturally occurring biomolecules vibrate. Labeling a specific biomolecule with a Raman tag allows it to be easily distinguished from the rest, increasing detection specificity. These tags can be classified into two main groups: isotope substitutions and tiny functional groups. Here, a brief explanation of the comparative advantages of these tags will be provided, but a more comprehensive analysis of Raman tags can be found elsewhere [54,55,56].

Isotope substitutions are one of the most straightforward modifications that can be introduced which causes the least perturbation of the properties of biomolecules. In Raman spectroscopy ^2^H or ^13^C isotopes are introduced in metabolic precursors to specifically tag the molecule of interest. Because the C-H bond is present in almost all biomolecules, isotope substitution can theoretically tag proteins, lipids, nucleotides, and carbohydrates. However, although they provide detection specificity with very low perturbation, the Raman signal of these small isotopic tags is very weak [54]. Thus, in practice, its use is usually limited to very abundant biomolecules that contain several C-H bonds, normally amino acids and lipids, that would be otherwise easily perturbed with bigger tags [57,58,59,60,61,62,63,64,65,66,67,68,69,70,71].

The second group of tags comprises tiny functional groups such as nitrile or alkyne moieties. These tags provide sharp and strong Raman peaks—a single nitrile group has a signal intensity equivalent to 11 C-^2^H bonds, and an alkyne more than double that [72]—yielding a large signal at low concentrations. Nevertheless, because of their bigger size, they can potentially introduce structural alterations to the biomolecules and require that the bioactivity of modified molecules be tested [55]. Nitrile tags have found several applications in biological systems [73,74,75,76], and most recently photoswitchable Raman probes with nitrile tags have been reported. In theory these opens the door for the development of super-resolution Raman microscopy techniques equivalent to the STORM or PALM approaches widely used in fluorescence microscopy [77,78]. Super-resolution Raman imaging has been long pursued, and although recently some interesting breakthroughs have come through [79,80,81,82,83,84], the development of new super-resolution approaches remains of high interest in the field.

Compared to nitriles, alkyne tags provide more than twice the signal intensity, and depending on the localization of the alkyne in the biomolecule, its Raman signal shifts—terminal alkynes appear at ≈2100 cm^−1^, and internal alkynes at ≈2200 cm^−1^—which allows for multiplexed imaging of different alkyne-tagged molecules if an appropriate combination of tag localization is selected [72]. Additionally, because of the recent popularity of click chemistry a huge variety of alkyne-tagged biomolecules are commercially available, which has been recently summarized by Bakthavatsalam et al. [56]. For these reasons alkyne tags are currently the most widely used Raman labeling strategy [54], and have been used to study lipids [57,72,85,86,87,88,89,90], proteins [85,86], sugars [86,91,92], and nucleotides [24,57,76,86,93,94].

The use of Raman tags in the study of the pathophysiological changes of brain diseases in vivo or their diagnosis is heavily limited. They require specialized delivery systems that allow crossing the blood–brain barrier and ensure their preferential incorporation into the lesion. However, recent exploratory attempts highlighted the feasibility of these Raman tags for studying pathophysiological changes in the brain [93], and drug delivery [95,96,97].

## 3. Raman Spectroscopy in Brain Pathophysiology

Raman spectroscopy has, in recent years, developed a growing interest in the study of diseases affecting the brain, because of the opportunity it provides for detecting biochemical changes with minimal invasiveness. This technique has been used not only for understanding the pathophysiology behind the diseases, but also for diagnosing the disease by detecting specific biomarkers. This section will discuss the application of Raman spectroscopy to the study and diagnosis of traumatic brain injury, ischemia, neurodegenerative diseases, and cancer. 

### 3.1. Traumatic Brain Injury and Ischemia

During a traumatic brain injury (TBI), two phases can be distinguished, primary and secondary. Primary brain injury comprises the pathophysiological changes derived from the trauma itself. In contrast, the changes derived from the host response to that trauma comprise the secondary brain injury phase, which plays a key role in determining the outcome of TBIs [98]. Thus, determining the changes occurring in the brain derived from a TBI is essential for completely understanding the process and developing effective diagnostic and therapeutic tools. The pathophysiology of secondary brain injury comprises a wide range of biological processes such as neuroinflammation [99], mitochondrial dysfunction [100], oxidative damage [101], and metabolic impairment [102], among others, which could serve as potential markers for the severity and prognosis of the injury. The main experimental techniques for sensing these chemical changes during TBI are microdialysis coupled with immunoassays, mass spectrometry, nuclear magnetic resonance, and, more recently, Raman spectroscopy [103]. The main comparative advantage of Raman spectroscopy is its potential in vivo application by using localized detection probes that allow for visualization of spatial changes in metabolite concentration. Additionally, this technique enables multiplex analysis of several biomolecules simultaneously, can be performed label-free, and is non-destructive. Nevertheless, Raman spectroscopy is not without limitations. Unlike other approaches, such as mass spectrometry, it cannot provide information about a specific protein or lipid species, in addition to the general limitations of the technique explored before.

The study of the metabolic changes during TBI by Raman spectroscopy started using mice as an animal model. In the first work of this type, mice were subjected to brain injuries and their whole brain was extracted. Then, these samples were excited with a 785 nm laser coupled to an optical microscope, and the Raman spectra were collected for 8 s. Authors found that injured brains showed a reduction of the amide I vibration at 1660 cm^−1^, accompanied by the apparition of sharp bands at 1560 and 1640 cm^−1^. Immunohistology revealed that these bands were associated with an increase in Caspase 3 levels and activation of apoptosis in neurons [104]. Other authors, also using whole mice brains as TBI models, were able to determine the temporal changes using confocal Raman microscopy. In the early “acute” phase, a strong signal associated to heme appeared due to initial hemorrhage. After 7 days, the heme signal disappeared, but an increase in the Raman band corresponding to cholesterol was observed, which was proposed to be related to cell repair processes [105]. Recently, the combination of Raman microscopy with Fourier transform infrared microspectroscopy in rat brain slices confirmed that cholesterol levels are elevated in the lesion site. This was accompanied by an increase in the Raman signal associated with proteins and a decrease in the phospholipid signal [106]. The same overall results were obtained in rat hippocampal slice cultures bio-orthogonally labeled with deuterium and alkyne tags. Using tagged analogues of DNA, RNA, proteins, and lipids, a 3-fold increase in protein and a 10-fold increase in palmitic acid levels were described, indicative of an activation of the cell repair mechanism [93].

Raman imaging of the whole brain or brain slices is a great tool to establish the metabolic changes occurring during TBI. Still, it is not a suitable approach for patient monitoring. For an efficient prognosis, low- or non-invasive techniques must be developed. In this regard, Raman spectroscopy was recently applied to the retina of mice TBI models. The spectra from the retina could distinguish between moderate and severe TBI, and chemical changes in the eye similar to those observed in the brain were recorded, associated with a release of cardiolipin after cell damage [107]. In another work, an artificial skull anatomical model was developed to guide the design of an intracranial probe prototype capable of delivering Raman measurements from clinically established cranial access techniques. This new device could measure real-time intracranial spectra while avoiding noise from extracranial tissue. Further validation of the device would allow it to be inserted at the lesion site and used to monitor changes after TBI during the acute phase [108]. The last set of approaches relies on measuring metabolic markers in biofluids for an easy and fast determination of the severity of the TBI. One of the main advantages of using biofluids such as blood or plasma is that they can be measured ex vivo by SERS, which greatly enhances the signal and sensitivity. SERS was applied to the detection of S-100β [109] or N-acetyl aspartate [110] as TBI biomarkers to determine the severity of the injury and potentially assist clinical decision-making. 

Cerebral ischemia results from insufficient blood flow to the brain to meet the metabolic demand. If the blood flow is reduced beyond a certain threshold, it can evolve into cerebral infarction, reaching a mortality rate of 15% [111,112]. During occlusion of cerebral blood flow, 1.9 million neurons can be lost per minute, so detecting cerebral ischemia in the early stages allows for treatment of the condition before it evolves, lowering the chance of infarction occurrence and improving survival rates [113,114]. Nowadays, the detection of cerebral ischemia relies primarily on magnetic resonance imaging (MRI) and computed tomography (CT). CT has a better overall resolution, is faster, more available, and less restrictive than MRI. Still, it requires the use of a contrast agent and radiation and is limited in the detection of small lesions [115]. Although both techniques constitute the gold standard for cerebral ischemia diagnosis, non-invasive, efficient, and economical complementary approaches are being developed to assist identification of the condition. Among these new approaches, Raman spectroscopy is a great candidate, as the reduction of blood flow and oxygen to a certain area of the brain produces metabolic changes than can potentially be detected by this technique.

In this regard, Raman spectroscopy has been used to measure cytochrome c (CytC) release in neurons as a marker for cell damage during cerebral ischemia. By measuring CytC release, Russo et al. demonstrated that insulin protects neurons during cerebral ischemia in a rodent animal model by modulating the oxidative stress resulting from restricted oxygen [116]. This work demonstrates that measuring CytC release by Raman is a feasible approach for detecting cerebral ischemia and determining its severity. Similarly, nitric oxide synthase (nNOS) inhibitors were found to protect the brain during ischemia. Using rat animal models, authors observed changes in the amide bands at 1276 and 1658 cm^−1^, and lipid bands at 1300 and 1438 cm^−1^ during ischemia, which were reverted when using nNOS inhibitors. The authors proposed that nNOS inhibition avoids the formation of reactive oxygen species, and the resulting vascular damage and apoptosis, protecting the brain [117]. More recently, Fan et al. measured biochemical changes in cerebral ischemia patients by Raman spectroscopy to develop a diagnostic tool. This non-invasive approach identified an increase in tyrosine, phenylalanine, and carotenoid levels after cerebral ischemia. Using machine learning, they developed a model that correctly classified the patients with accuracies approaching 100% using the spectra obtained from tear samples, demonstrating that Raman spectroscopy can be used for non-invasive screening of cerebral ischemia when several biomarkers are used in combination [118].

### 3.2. Neurodegenerative Diseases

Improvements in health care have resulted in a significant increase in lifespan in the past century [119], but this positive phenomenon has been unfortunately accompanied by an increase in the incidence of age-related neurodegenerative disorders, such as Parkinson’s disease (PD) and Alzheimer’s disease (AD) [120,121,122]. Many of these diseases are related to the apparition of protein aggregates, with other pathophysiological changes that are not yet completely understood. Importantly, these molecular changes can often predate the onset of clinical symptoms by decades. Thus, developing tools capable of detecting and characterizing the early molecular changes associated with the disease would serve a double purpose. First, it would allow for a complete understanding of the molecular basis of the disease and the rational design and development of new drugs. Second, it would allow for the application of these disease-preventing treatments before the apparition of neurodegenerative symptoms. To fulfill this second purpose, the pathophysiological changes associated with the disease must be detected early in patients, which requires identifying and characterizing suitable biomarkers. In this regard, Raman spectroscopy is a great candidate, as it provides the opportunity for non- or minimally-invasive biomarker detection approaches. This section will center on the use of Raman spectroscopy to identify and profile biomarkers for the diagnosis of neurodegenerative diseases. More information about the equally important use of Raman spectroscopy to characterize the molecular changes occurring during the development of neurodegenerative diseases can be found in the references [123,124,125,126,127]. 

The use of Raman spectroscopy for the identification of biomarkers and diagnosis ranges from minimally invasive to non-invasive methods. In the first set of approaches, the diagnosis of neurodegenerative diseases has been demonstrated in biopsies. Using SERS, León-Bejarano et al. measured α-synuclein in skin biopsies of PD patients. They found clear variations in the Raman bands of the protein, which shifted from 1655, 1664, and 1680 cm^−1^ to 1650, 1670, and 1687 cm^−1^, respectively, when comparing control and PD subjects. These changes were identified to be related to protein aggregation. This work demonstrates that SERS can detect the presence of aggregated α-synuclein in the skin, potentially allowing disease detection without the use of highly invasive tools [128]. Similarly, detection of α-synuclein aggregation has also been carried out in colon and olfactory bulb biopsies of Rat PD models, measuring protein phosphorylation as a marker of aggregation by Raman microspectroscopy [129]. The invasiveness of the detection method can be further reduced by using biofluids. In combination with infrared spectroscopy, Raman spectroscopy has been applied to blood plasma from PD patients and control individuals. A decrease in the bands associated with hydrocarbons (2990 cm^−1^), accompanied by an increase in the bands corresponding to amines (3200 cm^−1^) and alcohols (3300 cm^−1^) was observed in patients with the disease, probably derived from oxidative stress. Using these bands as biomarkers, the authors developed a discrimination model with 75% sensitivity and specificity [130]. The Raman profiling of platelets from mice [131] and rat [132] models of AD, as well as the profiling of blood plasma from human patients [133,134], also appear to be valid approaches for its detection and diagnosis, as several changes are observed in the platelet population and blood Raman profile during the development of AD. Blood samples have also been used to detect specific biomarkers of AD, such as the phosphorylated form of the p-tau protein. Using SERS, authors developed a diagnostic tool capable of detecting very low amounts of the protein, as low as 1.5 pg/mL, and accurately distinguishing AD patients from healthy subjects, as well as AD patients of different severities [135]. More recently, Bedoni et al. designed classification models for the diagnosis of Amyotrophic Lateral Sclerosis [136] and PD [137] by profiling the salivary fingerprint of patients using Raman spectroscopy. In both studies, the classification models presented accuracies, precision, sensibility, and sensitivity values above 97%, representing innovative non-invasive procedures for early detection of the diseases, with potential use in the future after further validation. This group also developed diagnosis models of PD [138] and AD [139] using the Raman profile of serum obtained from patients.

### 3.3. Cancer

Brain cancer and other nervous system cancers account for about 3% of all cancers diagnosed annually, although the incidence has been increasing over the last few years [140]. Brain cancer is one of the deadliest cancers worldwide, with a five-year relative survival rate of 35.8% [141]. It is considered the most intractable cancer type owing to the difficulty of completely removing it by surgery and the impediment that the blood–brain barrier represents for drug delivery [142]. Brain cancers are extraordinarily heterogeneous, encompassing more than one hundred types [143]; however, most can be classified as meningiomas and glioma tumors. Gliomas are more aggressive than meningiomas and affect glial cells that surround nerves and support the functioning of the central nervous system [144]. Astrocytomas are the most prevalent brain cancer in children, and glioblastomas in adults, both being glioma-type tumors [145]. Meningiomas, on the other hand, are often benign and arise from the meninges that surround the brain and the spinal cord [146]. 

A brain cancer diagnosis often involves a neurological exam, diverse imaging techniques (MRI, positron emission tomography (PET), or CT), and tissue biopsies. The combination of the information outcome from these studies allows the detection of the lesion’s area and the identification of the tumor type and grade. Although they are useful tools for diagnosis, they also present some disadvantages. First, the required equipment for imaging tests is expensive, which in some cases limits its utilization. Secondly, tissue biopsies imply surgery, an invasive procedure that entails risks associated to anesthesia and the surgery itself. Finally, the extracted tissue has to be processed and analyzed through histopathological techniques, which is laborious and time-consuming. The utilization of spectroscopic techniques, such as Raman spectroscopy, has overcome some of these limitations and emerged as a novel tool for brain cancer diagnosis [147]. Some of the main advantages of Raman spectroscopy over traditional methods are that: (i) it gives objective information concerning the biochemical composition of the sample; (ii) it can be less invasive since a variety of biological samples, such as biofluids, can be used, substantially reducing patient discomfort; and (iii) despite sample complexity, components presented in low concentrations are detectable. In addition, the combination of Raman spectroscopy or Raman imaging and machine learning techniques has recently reduced diagnostic time to a few seconds.

Since the first attempt to use Raman spectroscopy to differentiate healthy from edematous brain tissue [148], Raman spectroscopy has gained interest as a new diagnostic tool for brain tumors. Since then, several groups have delved into Raman’s ability for diagnosis, and even for grading and estimating the prognosis of brain cancers [147,149,150,151,152,153,154,155,156,157,158,159,160]. These studies highlighted distinct vibrational signatures for cancerous and healthy tissues. Reported differences in peak positions and/or intensities are due to quantitative or qualitative changes in biomolecules associated with the disease, and can therefore be used as biomarkers for cancer detection and, in some cases, categorization. Glioblastoma tissue exhibits significant differences from healthy tissue, mainly in areas corresponding to proteins and lipids [155,157,159]. The secondary structure of proteins shows divergences in α-helix and β-sheet content, while phosphatidylcholine and sphingomyelin’s characteristic peaks are shifted, which might be linked to molecular structural changes [157]. Moreover, bands corresponding to carotenoids, cytochrome c, and fatty acids have also been reported to be altered in human brain tumors, underlining their potential as Raman biomarkers [150,152,159]. Among fatty acids, it is worth highlighting the increase observed in linoleic acid in glioblastoma tissue, which is also accompanied by an increase in cholesterol levels [152]. This increase has been associated with the energy requirement of cancerous tissue [161] and an accumulation of lipid droplets [162], an organelle that serves as a reservoir mostly for triglycerides and cholesteryl esters. Raman spectroscopy has also revealed an increment in glioma tissue vascularization [152] and DNA and RNA content [150]. 

Raman spectroscopy can satisfactorily be used to diagnose a brain tumor using samples of very different nature, such as serum, freshly biopsied tissue, frozen tissue, patient-derived culture cells, and in vivo [150,158,160,163,164]. In addition to diagnosis, Raman spectroscopy can be a powerful tool to increase the precision of surgery. In the case of patients with glioma, neurosurgery is aimed at resecting the totality of the tumor, which is associated with a better prognosis and a reduction in cancer recurrence. However, total tumor resection is often unreachable due to infiltrative nature of cancerous cells. Against this background, the development of new technologies that allow for in vivo analysis of tissue in a surgical scenario seems indisputable. Raman spectroscopy has been shown to be valuable when identifying the tumor margins, which would facilitate total tumor removal [160,165]. Riva et al. have studied sixty-three fresh tissue glioma biopsies by pairing Raman spectroscopy with machine learning algorithms [160]. Cancerous and healthy brain tissues were distinguished ex vivo with an average accuracy of 83%. Other studies using glioma patient-derived cells have demonstrated that the spectral region going from 1000 cm^−1^ to 1300 cm^−1^ can discriminate between cancerous cells and healthy astrocytes with an average accuracy of 92.5% [150]. A surgical combination of tumor resection with in vivo Raman spectroscopy has shed promising results by accurately detecting tumor and infiltrated cancerous cells in real time [164,166]. The surgical application of Raman has been demonstrated to be highly sensitive (93%) and specific (91%) [164] and might be applied even for pediatric brain tumor resection [167].

## 4. Concluding Remarks

Brain affections, such as traumatic brain injury, ischemia, neurodegenerative disorders, or cancer, are diagnosed using a cohort of imaging and biochemical/histological techniques. Although effective for the diagnosis, and in some cases even for disease grading and prognostic assessment, these traditional techniques also have certain limitations. Most imaging techniques require expensive equipment that is not always available. In addition, tissue analysis requires extraction by brain biopsy, an invasive technique with associated risks for the patient. Raman spectroscopy and imaging overcome some of these limitations by providing a rapid and objective method based on the direct study of the chemical composition of label-free samples. Moreover, Raman techniques allow the utilization of various biological samples, including biofluids, which substantially reduces invasiveness in diagnosis. 

Improvement in Raman instrumentation has considerably increased spectral quality, allowing fine and reproducible detection of characteristic peaks and their changes in localization and intensity caused by brain diseases. Due to the effort of multiple research groups, many of the biochemical changes occurring during brain disease development and progression have been tracked by Raman spectroscopy and imaging, which has generated an array of well-established peaks and spectral regions that might be used for efficient clinical diagnoses. Additionally, Raman has been demonstrated to help in situ detection of lesions allowing selective resection. The increase in the interest of clinical applications is largely due to the combination of Raman spectroscopy/imaging with machine learning techniques, allowing for accurate discrimination between normal and disease-affected tissue, both ex vivo and in vivo. Moreover, combined utilization of Raman and machine learning techniques have reduced diagnostic time to a few seconds allowing real-time tissue analyses in a surgical scenario. Finally, recent research efforts have demonstrated that Raman tags can be used for quantifying specific biomolecules in pathophysiological processes, as well as for studying drug delivery to the brain both ex vivo and in vivo. This positions Raman as a potential candidate for understanding and diagnosing diseases and enhancing the efficiency of treatments by tracing drug delivery.

Altogether, in the last few years there has been significant progress in Raman instrumentation and automatized spectral analysis, highlighting a promising future for employing Raman-based in vivo diagnosis and treatment of brain ailments. 

## Figures and Tables

**Figure 1 ijms-24-02384-f001:**
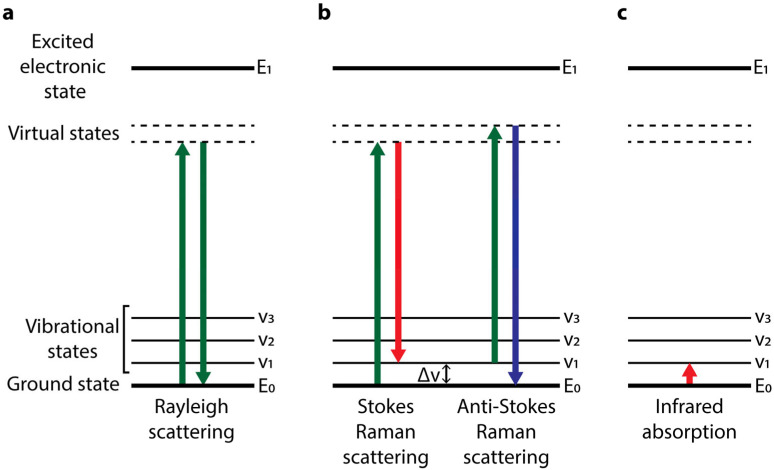
Energy level diagrams of Rayleigh scattering, Raman scattering, and infrared absorption. (**a**) In the elastic Rayleigh scattering, 1 out of 10^4^ photons that interact with matter is scattered with the same wavelength as the incident light. (**b**) In Raman scattering, when a molecule interacts with a photon, the vibrational energy of the molecule transitions to a higher virtual energy state, which is unstable and rapidly re-emits the photon. In the Stokes Raman scattering, the molecule starts at a ground energy level, and after scattering, it returns to a higher vibrational state, gaining energy in the process. The scattered photon has a lower frequency (and energy) than the incident light. In the anti-Stokes Raman scattering, the molecule starts in a higher vibrational energy state. After scattering, the molecule returns to the ground energy level, losing energy in the process. The scattered photon has a higher frequency (and energy) than the incident light. (**c**) In infrared absorption, the absorption of an incident photon induces the upward transition of the vibrational state of the molecule, and no photon is re-emitted.

**Figure 2 ijms-24-02384-f002:**
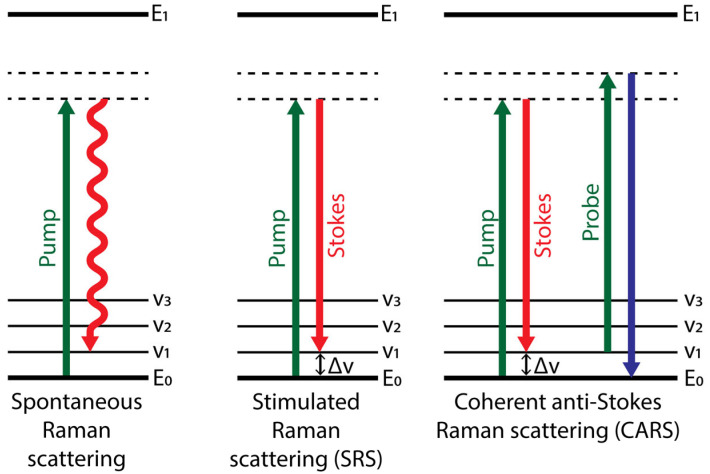
Energy level diagrams of different modes of Raman spectroscopy and imaging. Spontaneous Raman scattering irradiates the target molecule with a pump laser (green). In 1 out of 10^8^ interactions between the incident photons and the matter, the photon induces the transition of the molecule’s vibrational energy level to a higher virtual state. Then, the photon is spontaneously re-emitted in the scattering event, usually with a lower frequency. In Stimulated Raman Scattering (SRS), the target molecule is irradiated with a pump laser (green). Then, a second Stokes laser (red) is used so that the frequency difference between the two lasers (Δv) resembles that of a specific vibrational transition, enhancing its occurrence. In Coherent anti-Stokes Raman Scattering (CARS), the target molecule is irradiated with pump and Stokes lasers, as in SRS, to induce a specific vibrational transition, followed by a third probe laser, inducing the return of the molecules vibrational state to the ground level and the re-emission of a photon with higher frequency.

## Data Availability

The data presented in this study are available in article.

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
