# Peer review of "Raman Spectroscopy as a Tool to Study the Pathophysiology of Brain Diseases"

_ijms, 2023, doi:10.3390/ijms24032384_

Round 1
Reviewer 1 Report
The authors present a nice review about Raman spectroscopy and its applications by enumerating a large number of literatures published in the last few years. They start from the basics of Raman spectroscopy, then turn to Raman spectroscopy, Raman modes and Raman tags, finally move to the topic of application of Raman techniques in the pathophysiology of brain diseases. I think that this work deserves the publication in the international journal of molecular sciences. However, the following points have to be addressed before the publication.
1. Could the authors clarify “some of the problems” additionally in the lines 102-103 ?
2. An important Raman spectral region, low wavenumber region or Terahertz region, has been skipped by the authors. Actually, this regime contains rich information, such as the delocalized vibrational modes of macromolecules (Biophysics Journal, 48, 289, 1985; Proc. Natl. Acad. Sci. USA 82, 4995,1985; Nature Communications, 5, 3999, 2014). These vibrational modes are believed to have important biological functions. In the low-wavenumber region, a significant progress has been made recently for the coherent Raman scattering techniques (Optics Letters, 47, 22, 2022; ACS Photonics, 7, 3481, 2020; Optica, 6, 52, 2019; Optics Letters, 43, 470, 2018). It would be more comprehensive for a review paper if the authors can fill this spectral gap.
3. The authors must double check the typing errors in the context. For example, there is a typing error in the line 382.
Author Response
We thank reviewers for their critical comments and important suggestions that helped us to improve our manuscript significantly. Any changes made to the main manuscript file have been introduced with “Track Changes” in the revised manuscript.
Reviewer #1:
The authors present a nice review about Raman spectroscopy and its applications by enumerating a large number of literatures published in the last few years. They start from the basics of Raman spectroscopy, then turn to Raman spectroscopy, Raman modes and Raman tags, finally move to the topic of application of Raman techniques in the pathophysiology of brain diseases. I think that this work deserves the publication in the international journal of molecular sciences. However, the following points have to be addressed before the publication.
We thank reviewer 1 for their positive comments.
- Could the authors clarify “some of the problems” additionally in the lines 102-103?
The original sentence, “[Raman spectroscopy] overcomes some of the problems associated with IR spectroscopy,” refers to the low sample penetration and spatial resolution of IR spectroscopy mentioned in the previous sentence. To further clarify this point, the sentence in line 106 has been changed to read, “Comparatively, Raman spectroscopy uses near-infrared or visible light, which provides higher sample penetration and spatial resolution.” See track changes in the revised manuscript.
- An important Raman spectral region, low wavenumber region or Terahertz region, has been skipped by the authors. Actually, this regime contains rich information, such as the delocalized vibrational modes of macromolecules (Biophysics Journal, 48, 289, 1985; Proc. Natl. Acad. Sci. USA 82, 4995,1985; Nature Communications, 5, 3999, 2014). These vibrational modes are believed to have important biological functions. In the low-wavenumber region, a significant progress has been made recently for the coherent Raman scattering techniques (Optics Letters, 47, 22, 2022; ACS Photonics, 7, 3481, 2020; Optica, 6, 52, 2019; Optics Letters, 43, 470, 2018). It would be more comprehensive for a review paper if the authors can fill this spectral gap.
We thank reviewer 1 for their comments. We agree that the low-frequency region provides valuable information concerning the structural conformation of biomolecules and should be mentioned in our manuscript. We have thus followed reviewer 1’s recommendation and added a new paragraph (lines 130-140 of the corrected manuscript), explaining the characteristics of this spectral region and also mentioning some limitations that have restricted the utilization of Raman spectroscopy and microscopy of low-frequencies. We have also introduced some of the references suggested by reviewer 1, together with others we found interesting.
We believe that the description of the low-frequency region has improved the quality and relevance of our review paper, and we thank reviewer 1 again for their comments.
- The authors must double check the typing errors in the context. For example, there is a typing error in the line 382.
We have checked the manuscript for typing errors as suggested by reviewer 1.
Reviewer 2 Report
The authors propose the manuscript entitled: Raman spectroscopy as a tool to study the pathophysiology of brain diseases as a review article in IJMS (6.208 for 2021). This journal is one of highest rang of MDPI.
The manuscript is very detailed and despite that I did not find any serious deficiency or error. I have only few minor comments:
1) From my point of view, the first part (1. Raman scattering) is not referenced sufficiently. The whole section is covered by the origin CV Raman articles from 1928 (In fact, Raman and Krishan just described existence of the “new type of radiation” in this articles). I highly recommend to the authors supplemented some serios references to this section. I highly recommend to the authors complete some references to this section. For inspiration, there is some possibilities:
ISBN: 978-0-471-49028-9
ISBN:978-0-12-254105-6
DOI:10.1002/0470011831
https://doi.org/10.1016/C2009-0-21628-X
Feel free to use completely difference references, but the only one reference from 1928 are not able to cover such wide topic as Raman spectroscopy in the year 2023.
2) Please change or re-write sentence on the line 126-127:
The first is the “fingerprint region” which ranges from 600 cm-1 to 1800 cm-1 and comprises the most important region for chemical identification. This region has a prominent contribution of signals derived mainly from proteins and nucleic acids.
Generally, it is not true. Also, Raman signatures of carbohydrates, lipids etc. can be found there.
3) Citation from lines 121-123:
Virtually all kinds of biomolecules contribute to the Raman spectra of biological samples. Since different biomolecules have specific chemical signatures, several unique vibrational bands can be used for their identification.
These strong claims also deserve citation, for inspiration:
10.3390/s7081343
10.1140/epjp/s13360-021-01152-1
10.1002/cphc.200390004
10.1364/AO.42.002724
As my previous recommendation, feel free to use another reference.
Anyway, I see these as minor flaws and I recommend the editor to accept the manuscript as a review article in IJMS.
Author Response
We thank reviewers for their critical comments and important suggestions that helped us to improve our manuscript significantly. Any changes made to the main manuscript file have been introduced with “Track Changes” in the revised manuscript.
Reviewer #2:
The authors propose the manuscript entitled: Raman spectroscopy as a tool to study the pathophysiology of brain diseases as a review article in IJMS (6.208 for 2021). This journal is one of highest rang of MDPI.
The manuscript is very detailed and despite that I did not find any serious deficiency or error.
We thank reviewer 2 for their positive comments.
I have only few minor comments:
1) From my point of view, the first part (1. Raman scattering) is not referenced sufficiently. The whole section is covered by the origin CV Raman articles from 1928 (In fact, Raman and Krishan just described existence of the “new type of radiation” in this articles). I highly recommend to the authors supplemented some serios references to this section. I highly recommend to the authors complete some references to this section. For inspiration, there is some possibilities:
ISBN: 978-0-471-49028-9
ISBN:978-0-12-254105-6
DOI:10.1002/0470011831
https://doi.org/10.1016/C2009-0-21628-X
Feel free to use completely difference references, but the only one reference from 1928 are not able to cover such wide topic as Raman spectroscopy in the year 2023.
We agree with Reviewer 2 on this point. When introducing the Raman scattering, we wanted to make sure that we credited the original discovery of the phenomenon. Nevertheless, as reviewer 2 points out, the original paper by Raman and Krishan does not delve into the physical explanation or characterization of the inelastic scattering. Thus, the current understanding of the phenomenon is built on posterior works by other authors, collected in several books and reviews that must also be cited. This has been addressed with a new sentence in lines 41-43 of the revised manuscript and the introduction of new references as suggested by reviewer 2 (lines 64 and 69). See track changes in the revised manuscript.
2) Please change or re-write sentence on the line 126-127:
The first is the “fingerprint region” which ranges from 600 cm-1 to 1800 cm-1 and comprises the most important region for chemical identification. This region has a prominent contribution of signals derived mainly from proteins and nucleic acids.
Generally, it is not true. Also, Raman signatures of carbohydrates, lipids etc. can be found there.
We thank reviewer 2 for their correction. Indeed, the fingerprint region comprises localized spectral features of a wide variety of biomolecules, not only proteins and nucleic acids. The sentence has been corrected (lines 143-144 of the corrected manuscript), and additional information about lipids and carbohydrates has been added (lines 154-162 in the corrected manuscript).
3) Citation from lines 121-123:
Virtually all kinds of biomolecules contribute to the Raman spectra of biological samples. Since different biomolecules have specific chemical signatures, several unique vibrational bands can be used for their identification.
These strong claims also deserve citation, for inspiration:
10.3390/s7081343
10.1140/epjp/s13360-021-01152-1
10.1002/cphc.200390004
10.1364/AO.42.002724
As my previous recommendation, feel free to use another reference.
The sentence was added as an introduction for a more detailed description of the different regions that follows, where several works are cited. Nevertheless, we agree that providing certain references in that sentence would aid the reader. We have followed reviewers 2 recommendation and added new references in line 128 of the revised manuscript.
Anyway, I see these as minor flaws and I recommend the editor to accept the manuscript as a review article in IJMS.
We thank reviewer 2 again for their positive comments.